# Specific Gene Duplication and Loss of Cytochrome P450 in Families 1-3 in Carnivora (Mammalia, Laurasiatheria)

**DOI:** 10.3390/ani12202821

**Published:** 2022-10-18

**Authors:** Mitsuki Kondo, Yoshinori Ikenaka, Shouta M. M. Nakayama, Yusuke K. Kawai, Mayumi Ishizuka

**Affiliations:** 1Laboratory of Toxicology, Department of Environmental Veterinary Science, Faculty of Veterinary Medicine, Hokkaido University, Sapporo 060-0818, Japan; 2Water Research Group, Unit for Environmental Sciences and Management, North-West University, Potchefstroom 2520, South Africa; 3Translational Research Unit, Veterinary Teaching Hospital, Faculty of Veterinary Medicine, Hokkaido University, Sapporo 060-0818, Japan; 4One Health Research Center, Hokkaido University, Sapporo 060-0818, Japan; 5Biomedical Sciences Department, School of Veterinary Medicine, The University of Zambia, P.O. Box 32379, Lusaka 10101, Zambia; 6Laboratory of Toxicology, Department of Veterinary Medicine, Obihiro University of Agriculture and Veterinary Medicine, Obihiro 080-8555, Japan

**Keywords:** cytochrome P450, genome database, in silico analysis, phase I metabolism, wildlife, xenobiotic metabolism

## Abstract

**Simple Summary:**

In this study we investigated the specific duplication and loss events of cytochrome P450 (CYP) genes in families 1-3 in Carnivora. These genes have been recognized as essential detoxification enzymes, and, using genomic data, we demonstrated a synteny analysis of the CYP coding cluster and a phylogenetic analysis of these genes. We discovered the CYP2Cs and CYP3As expansion in omnivorous species such as the badger, the brown bear, the black bear, and the dog. Furthermore, phylogenetic analysis revealed the evolution of CYP2Cs and 3As in Carnivora. These findings are essential for the appropriate estimation of pharmacokinetics or toxicokinetic in wild carnivorans.

**Abstract:**

Cytochrome P450s are among the most important xenobiotic metabolism enzymes that catalyze the metabolism of a wide range of chemicals. Through duplication and loss events, CYPs have created their original feature of detoxification in each mammal. We performed a comprehensive genomic analysis to reveal the evolutionary features of the main xenobiotic metabolizing family: the CYP1-3 families in Carnivora. We found specific gene expansion of CYP2Cs and CYP3As in omnivorous animals, such as the brown bear, the black bear, the dog, and the badger, revealing their daily phytochemical intake as providing the causes of their evolutionary adaptation. Further phylogenetic analysis of CYP2Cs revealed Carnivora CYP2Cs were divided into CYP2C21, 2C41, and 2C23 orthologs. Additionally, CYP3As phylogeny also revealed the 3As’ evolution was completely different to that of the Caniformia and Feliformia taxa. These studies provide us with fundamental genetic and evolutionary information on CYPs in Carnivora, which is essential for the appropriate interpretation and extrapolation of pharmacokinetics or toxicokinetic data from experimental mammals to wild Carnivora.

## 1. Introduction

Cytochrome P450s (CYP) catalyze major xenobiotics metabolism for a wide range of chemicals, such as drugs, phytochemicals, and environmental pollutants, and are considered the most important enzymes for detoxification [1]. The CYP genes form a superfamily and are divided into gene families based on >40% sequence similarity, but into subfamilies when sequence similarities are >55%. Among various CYP families in mammals, the CYP1-3 families are considered one of the main families that catalyze xenobiotic metabolism [2,3,4], although other CYP families are generally considered important for the biosynthesis of numerous endogenous chemicals such as steroids, bile acids, cholesterols, eicosanoids, fatty acids, etc. [5,6].

These CYP genes are considered the fastest-evolving gene systems [7,8,9] and through gene duplication and loss events, xenobiotic-metabolizing CYPs, CYP1-3 genes, have evolved, duplicated or been lost, and diversified. Such an evolutionary history is essential for characterizing isoform-specific substrate-specificity and further characterizing species-specific metabolism in animals [10]. Reports suggest that CYP genetic duplication or loss affects the xenobiotic metabolism capacity in several animals and that these evolutionary consequences might be due to the need of some insects, birds, and herbivorous mammals to manage constant exposure to phytochemicals [11,12,13,14]. This “plant-arms race” concept may explain the necessity of CYP duplication events in herbivorous species: it hones detoxification systems in response to phytochemicals or toxins originating in plants. However, a detailed analysis of CYP duplication events in species of the Carnivora order is not available. Previous reports of CYP2D subfamily in mammals showed no clear relation with CYP2D expansion or contractions and food habitat in mammals, but suggested that some isoforms did not necessarily follow this “plant-arms race” hypothesis [15]. However, this study included only one isoform subfamily with a wide range of mammals, and evaluation for CYP expansion or contraction in various isoform subfamilies is still needed. 

Carnivora is a mammalian order consisting of 297 species, including cat-like (Feliformia) and dog-like (Caniformia) animals [16,17]. As its name suggests, several animals within this order are largely carnivorous, such as Felids, yet foraging habits of the different species are diverse in this group. Previous research has identified genetic loss of certain xenobiotic metabolism enzymes. UDP-glucuronosyltransferase (UGT) in felids and pinnipeds indicates a strong relationship between carnivory and genetic loss of xenobiotic metabolism enzymes [18,19,20]. However, there has been little research investigating CYP genetic loss and duplication events in this taxon, leading us to comprehensively investigate their evolutionary history in Carnivora species and the relationships between foraging habits and their genetic duplication or loss events.

Recent innovations in next generation sequencing (NGS) systems have enabled us to manage large amounts of genetic data from a wide range of wild mammals, and we can also utilize these high-quality genetic assembly data freely through an online database [21]. These whole genomics data enabled us to comprehensively analyze genetic duplication and loss events in Carnivora, and further to compare phylogenetic relationships of each gene to provide an understanding of the CYP evolutionary features in this taxon. 

Firstly, we investigated and compared the synteny of CYP1-3 gene isoforms’ loci in Carnivora species. We then conducted a phylogenetic analysis to further detected specific gene duplication or loss events in each studied species to reveal the evolutionary features of CYP genes in Carnivora.

## 2. Materials and Methods

### 2.1. Data Retrieval for CYP Phylogenetic Analysis

Since this study does not use samples from living organisms, no ethics committee approval review of animal experiments is required. Phylogenetic analyses were performed on the CYP genes of human (*Homo sapiens*), dog (*Canis lupus familiaris*), red fox (*Vulpes vulpes*), domestic ferret (*Mustela putorius furo*), ermine (*Mustela erminea*), mink (*Neovison vison*), badger (*Meles meles*), North American river otter (*Lontra canadensis*), Eurasian river otter (*Lutra lutra*), sea otter (*Enhydra lutris kenyoni*), polar bear (*Ursus maritimus*), giant panda (*Ailuropoda melanoleuca*), black bear (*Ursus americanus*), brown bear (*Ursus arctos*), meerkat (*Suricata suricatta*), striped hyena (*Hyaena hyaena*), domestic cat (*Felis catus*), Amur tiger (*Panthera tigris*), cheetah (*Acinonyx jubatus*), puma (*Puma concolor*), Canada lynx (*Lynx canadensis*), leopard (*Panthera pardus*), lion (*Panthera leo*), leopard cat (*Prionailurus bengalensis*), fishing cat (*Prionailurus viverrinus*), Weddell seal (*Leptonychotes weddellii*), harbor seal (*Phoca vitulina*), gray seal (*Halichoerus grypus*), Hawaiian monk seal (*Neomonachus schauinslandi*), northern elephant seal (*Mirounga angustirostris*), southern elephant seal (*Mirounga leonine*), northern fur seal (*Callorhinus ursinus*), Steller’s sea lion (*Eumetopias jubatus*), California sea lion (*Zalophus californianus*), and Pacific walrus (*Odobenus rosmarus divergens*). Sequences were retrieved using National Center for Biotechnology Information (NCBI) BLAST searches using the following query sequences: human and dog CYP1A1, 1A2, and 1B1 for CYP1 searching, human CYP2A6, 2B6 2D6, 2E1, 2F1, 2J2, 2S1, 2U1, 2W1, 2S1, rat 2T1 and 2G1, and dog 2C21, 2C41, and 2C23 were used for CYP2 investigation, and human CYP3A4, dog CYP3A12 and 3A26, and cat CYP3A131 and 132 were used for CYP3. These isoform queries were sufficiently comprehensive for detecting target genes and hitting other additional subfamily isoforms in Carnivora (e.g., the CYP2C6 BLAST search also detected CYP2Es and other subfamily genes). BLAST searches were conducted on the database Nucleotide collection (nr/nt) for each species using BLASTN (optimized for similar sequences). The gene sequences used are listed in the Appendix A, and the protein coding region of each isozyme was analyzed. The deduced amino acid sequences were aligned using MUSCLE (Multiple Sequence Comparison by Log-Expectation) and were used for model selection (models showing minimal sets of BIC were chosen) and construction of maximum likelihood trees (bootstrapping = 100) using MEGA X (Molecular Evolutionary Genetics Analysis) [22]. The aligned sequence lengths analyzed were 1965 bp in CYP3As and 1581 bp in CYP2Cs. The JTT+G model was used. All positions containing gaps and missing data were eliminated, and the total lengths of protein-coding sequence alignments were used for phylogenetic analysis. Foraging habits of each analyzed species are also listed in the Appendix A. The phylogenetic tree was created with TimeTree 5 [23]. We also provided protein sequence files in Supplementary Information: CYP3A_protein.fas, and CYP2C_protein.fas.

### 2.2. Synteny Analysis of CYP Genes

Sequence data from genome projects are freely available. NCBI’s genome data viewer (https://www.ncbi.nlm.nih.gov/genome/gdv/ (accessed on 1 October 2022)) or JBrowse [24] were used to visualize the chromosomal synteny maps for each species. The following latest genome assemblies were used and listed in Appendix A. UCSC (University of California, Santa Cruz) BLAT (a BLAST-like alignment tool) (http://genome.ucsc.edu/index.html (accessed on 1 October 2022)) was used for additional confirmation of missing genes. The masked palm civet CYP genes were also retrieved and used to fill the gap for Feliformia species from recently assembled and annotated chromosome-level genomic data [25]

## 3. Results

### 3.1. CYP Number Counts and Isoforms in CYP1As and 2ABGFSs Clusters

Gene number counts for CYP 1-3 genes are shown in Figure 1**.** We found CYP gene coding loci where multiple CYPs were coded as a “cluster” of the CYP genes. CYP1As, CYP2ABGFSs, CYP2Cs, and CYP3As in each Carnivoran consist of a gene cluster. Several CYP gene clusters were conserved among Carnivorans. The CYP2ABFGSTs cluster coded CYP2As, 2Bs, 2Fs, 2Gs, and 2Ss annotated genes and was between *AXL Receptor Tyrosine Kinase* (AXL) and *Egl-9 family hypoxia inducible factor 2* (EGNL2), and the CYP1As cluster was between *C-terminal Src kinase* (CSK) and *Enhancer of mRNA decapping 3* (EDC3) when CYP1A1 and 1A2 orthologs in analyzed Carnivora were coded in this cluster. We further found specific gene duplication in brown bear CYP2As in the CYP2ABFGSTs cluster (Figure 1).

### 3.2. CYP Isoforms in CYP2Cs and CYP 2CEs Clusters

Synteny of CYP2C coding loci are shown in Figure 2. The CYP2Cs coding loci also consisted of gene clusters, labeled the CYP2Cs clusters, and were highly conserved between *Helicase, lymphoid specific* (HELLS) and the *PDZ and LIM domain 1* (PDLIM1) among carnivorans, humans, and rodents. Multiple CYP2Cs were coded in the Carnivoran cluster. In almost all Mustelidae (except badger), Felidae (except the domestic cat), Pinnipedian, Canidae, and meerkat, the analyzed genome had two CYP2Cs annotated genes in this cluster, whereas CYP2Cs in badger had three isoforms (with one isoform in another un-scaffolded contig), and the domestic cat had one intact isoform and one possible dysfunctional gene. The striped hyena genome had three isoforms in this cluster, whereas the meerkat had two. Some species genomes had CYP2Cs in several un-scaffolded contigs, and we did not find completely connected cluster loci for the Pacific walrus genome although we found two possible isoforms in different contigs. 

In contrast, within Ursidae we found huge species differences. The giant panda genome contained one possible CYP2C in this cluster and additional un-scaffolded isoforms in the contig NW_023254381.1. In contrast, the polar bear genome contained two possible CYP2Cs in this cluster with a partial additional isoform (CYP2C41-like) in the un-scaffolded contig NW_024425153.1. However, in the brown bear genome, we found three annotated CYP2Cs in this cluster. Moreover, the black bear genome had five possible isoforms and three partial isoforms, even though this cluster seemed to be on two separated contigs, and the partial isoforms on two other contigs (Figure 2).

We further found other specific CYP2Cs loci coding CYP2Cs and 2Es in Carnivora between *Synaptonemal complex central element protein 1* (SYCE1) and *Scavenger receptor family member expressed on T Cells* 1 (SCART1): which were labeled the CYP2CEs cluster. Only Canids, ursids, and Pinnipeds had CYP2Cs in this cluster whereas other Carnivoran genomes (Feliformia (Felidae, striped hyena, and meerkat) and Mustelidae) had only CYP2Es in this cluster.

Other CYP2 subfamily genes such as CYP2Ds, 2Js, 2Rs, 2Us, and 2Ws did not show any duplication in Carnivora coded as isolated genes, although coding loci were highly conserved among all analyzed Carnivorans and among other mammals (data not shown). Nevertheless CYP2J, 2R, 2S, 2T, 2U, and 2Ws are generally known as biosynthesis-type or unknown substrate isoforms, and we did not include these genes in the analyses.

### 3.3. Synteny Analysis of the CYP3As Cluster

We also analyzed the CYP3As gene cluster shown in Figure 3, and this cluster, which was also conserved among carnivorans, is between *Zinc finger and SCAN domain containing 25* (ZSCAN25) and *Olfactory receptor family 2 subfamily AE member 1* (OR2AE1) or *Tripartite motif containing 4* (TRIM4). All Felidae analyzed had only two possible isoforms of CYP3As, and no species-specific differences were observed. However, the Mustelidae genome had four or three possible isoforms in the CYP3As cluster and the ermine and Canadian river otter had one pseudogene-annotated gene in this cluster. In Ursidae, the brown and black bear genomes also contained four annotated CYP3As, whereas the polar bear and black bear genomes had three isoforms. The giant panda genome contained only one isoform annotated as “LOW-QUALITY PROTEIN” coding gene in this cluster, with six other very short partial un-scaffolded isoforms (CDS: CoDing Sequence length less than 515 bp) observed. Canids also have multiple CYP3As in this cluster, and the dog genome had four isoforms in this cluster (chromosome 6: NC_051810.1) with several intact and partial isoforms. Recently, two dog CYP3As were characterized and renamed, and the NCBI annotated name was different to the CYP nomenclature in dog CYP3As. The NCBI naming system was followed and genes were renamed in this paper (CYP3A4; Gene ID: 479740 as CYP3A98 and CYP3A12-like; LOC119875773 as CYP3A99) [26]. Red fox CYP3As were also on three un-scaffolded contigs (NW_020356965.1, NW_020356599.1, and NW_020356653.1) with two isoforms and three partial isoforms. However, in the Arctic fox genome, CYP3As were not coded as a cluster and these genes were coded on two different loci on the same chromosome (chromosome 3: NC_054826.1 with two intact and one short isoform). In several Pinnipedia genomes, CYP3As were also located on several un-scaffolded contigs, suggesting much higher quality assemblies are essential for clear analysis. Three Otariidae or Odobenidae genomes (from the Pacific walrus, northern fur seal, and Stellar sea lion) had two intact or partial CYP3As and one annotated pseudogene of CYP3As, whereas four isoforms were observed in the California sea lion genome (Appendix A). Phocidae genomes also have scattered genes of CYP3As and we could not find a clear CYP3A cluster or isoform, with 1–2 intact CYP3As and several partial genes in each genome. 

### 3.4. Phylogeny of CYP2Cs in Carnivorans

We performed a phylogenetic analysis of CYP2Cs in carnivorans, and we revealed Carnivoran CYP2Cs were divided into three clades, namely CYP2C41s, CYP2C21s, and CYP2C23s (Figure 4). Based on the phylogeny, the CYP2C23s clade was located close to CYP2Es clades. Each clade contained orthologous Carnivoran genes to CYP2C41 and 2C21 in dogs and CYP2C23 in rats, respectively. Almost all CYP phylogenies in the CYP2C41s, 2C21s, and 2C23s clades followed their organisms’ phylogeny order. We also found specific duplication of the CYP2C21s clade in Ursidae, suggesting these duplication events occurred after the divergence of Ursidae.

### 3.5. Phylogeny of CYP3As in Carnivorans

We conducted similar phylogenetic analysis on CYP3As that revealed a Caniformia-clade and a Feliformia-unique clade in CYP3As (Figure 5), suggesting the CYP3As evolutionary history between Feliformia and Caniformia was completely different. Based on the phylogeny, the Caniformia-clade was further subdivided into three clades which were labeled Caniformia CYP3As clade 1 to clade 3. For clade 1, almost all Caniformia species possessed these genes, and Mustelidae had two each specifically duplicated clades (1-1 and 1–2), suggesting canid-specific and Mustelid-specific duplication of genes in this clade. Canidae also showed lineage specific duplication in this clade. Ursidae, Mustelidae, and some Pinniped genomes possessed CYP3As in clade 2, whereas Canidae and Feliformia did not have isoforms in this clade. In clade 3, however, Canidae, Mustelidae, and Ursidae have these genes and canids have a family-specific duplication of this clade similar to clade 1. Although some pinnipeds have multiple CYP3As, some with partial genes were removed for phylogenetic analysis to ensure clear results were produced, so there could be some pinniped CYP3As that might have been classified into clade 3. 

For the CYP3As Feliformia-clade, Felidae CYP3As were divided into two clades named CYP3A131s and CYP3A132s as per the domestic cat CYP3A131 and CYP3A132 [27]. In each clade, specific clades for Felidae were further established and only one isoform from each respective Felidae species was contained in these clades. However, other Feliformia CYP3As genes were not classified into these CYP3A131s and CYP3A132s clades, suggesting unique CYP3As loss or duplication events occurred in each species through the evolutionary history of Feliformia.

## 4. Discussion

In this analysis, we utilized the genome assembly data for multiple Carnivorans, and still these data have several limitations. Genome-assembled data only represent one individual element of genomic information and several variants could happen in all genomic assembly. Also, genome-assembly qualities should be considered. Although recent genomic assemble quality has improved, and several chromosomal-level assemblies were available even for wildlife Carnivorans, some assemblies have relatively low quality which made it difficult for us to analyze some of the CYP genes. However, we considered this genomic data analysis to have strong importance in the evaluation of the wildlife CYP genome. 

### 4.1. CYPs Duplication and Loss in Mammals and Relationships with Foraging Habits

Several reports have revealed CYP gene duplication events in a variety of mammals, especially herbivorous mammals. Recently, a koala genome project revealed that a huge expansion of CYP2Cs was possibly an adaptation to a diet of eucalypts [11]. Further, among woodrat genomes, especially juniper-eating species, several reports have found higher gene copy numbers of CYP2As, 2Bs, and 3As compared to other rodents [12,28,29], and studies have also suggested that the woodrat’s CYP2Bs gene expansion might contribute to their high metabolic capacity for terpenes from juniper plants. In this study, we found specific duplication of CYP2Cs and CYP2As in the brown bear, CYP2Cs in the black bear, CYP2Cs and 3As in the badger, and CYP3As in the dog. All these animals are omnivorous and their foraging habits include a wide variety of food types, indicating that these species have a “generalist” diet [30,31,32,33,34]. Therefore, these gene expansions might be the consequences of a need for detoxification of a wide variety of plant-secondary metabolites in their daily diets. 

However, in other omnivorous or herbivorous animals, (e.g., the red fox, Arctic fox, and giant panda), we did not find any specific duplication of CYPs. In the red fox genome, CYP3As clusters were separated because of assembly quality, which suggested that these regions require re-assembling or target re-sequencing. Further, we found separated clusters of CYP3As for the Arctic fox, suggesting that these loci in Arctic fox genome are instable. Hence, further in-depth genomic analysis is required to clarify whether the CYP3A expansion is limited to dogs or is also applicable to other Canidae species.

Notably, we assumed giant panda genomes would show expansion of CYPs in response to various plant secondary metabolites in their exclusively bamboo diets. However, our result suggested that there was neither expansion of any CYPs, nor even a contraction trend of CYP3As in the giant panda. These trends were supported by our analysis of UGTs genomics for this species (unpublished data). These results strongly suggested that giant panda do not rely on an “enzymatic-strategy” to deal with their daily toxin intake. Their unbalanced “specialist” bamboo-exclusive diet might be the reason they have not evolved or expanded CYP or other xenobiotic metabolism enzymes, and gut microbiota is an alternative strategy they might use instead [35,36,37]. 

### 4.2. CYP2Cs in Carnivora

From the phylogenetic analysis of this study, we identified that Carnivora CYP2Cs are divided into three clades, which were possibly orthologs of CYP2C21s, CYP2C41 in the dog, and CYP2C23s in the rat [38,39,40]. These features strongly suggest that the substrate specificity of dog CYP2Cs is similar to that of CYP2C21 and CYP2C41 in other Carnivora. Previous studies have revealed that dog CYP2C21 showed substrate specificity for diclofenac, midazolam, and the 4-methyl-N-methyl analog of sulfaphenazole, O-desmethyltramadol, testosterone (16-alpha OH), and (S)-Mephenytoin [39,41,42,43]. CYP2C41 also showed similar substrate specificity to diclofenac and midazolam albeit with lower activity. However, comprehensive and systematic analysis of recombinant CYP2C21 and CYP2C41 in the dog is necessary for the clarification of specific substrate-specificity. Our results together with this substrate specificity suggest that Carnivora CYP2C21s and 2C41s might also show similar substrate specificity. Our analysis revealed brown bear and black bear also showed gene expansion in CYP2C21 and 2C41s. Since data regarding substrate specificity of CYP2Cs in Carnivora are limited, we need to estimate Carnivora CYP2Cs in Ursidae from other mammals. Human CYP2Cs showed metabolism of a wide variety of chemicals including endogenous eicosanoids and fatty acid [44], xenobiotic drugs such as antimalarials, oral antidiabetics, most NSAIDs, most proton pump inhibitors and warfarin [2,45], and some terpenoids [46,47,48]. Interestingly, brown bears, black bears, and even badger are known to consume pine nuts [33,49,50], which are from conifer trees that contain terpenoids [51,52]. Further study investigating recombinant CYP2Cs is essential to confirm whether these expanded CYP2Cs in the Ursid and badger are able to metabolize the terpenoids in their daily diets. 

Among Canids, CYP2C41s show polymorphism as a complete loss in several breeds [53], suggesting some dogs have contracted CYP2Cs. This further suggests that other species might also show polymorphism or copy number variants within species, providing further justification for the importance of genomic analyses that use several individuals to conclude isoform numbers in each species. 

### 4.3. CYP2C23s Orthologous Genes in Carnivora

From the phylogenetic analysis, we discovered a possible orthologue to rat CYP2C23 and mouse CYP2C44 in Carnivora. The coding locus of rat CYP2C23 and mouse CYP2C44 (between Erlin1 and Cpn1) was neither in the CYP2Cs cluster, nor in a similar region to that of Carnivorans. These isoforms have been cloned and characterized as arachidonic acid or eicosanoid metabolizing CYPs and are closely related to the endogenous biosynthesis of these animals [38,54,55], although humans have pseudogenes of these isoforms. We found possible orthologues to rodents CYP2C23s in Carnivora, which strongly suggests these isoforms in Carnivora also have similar eicosanoid metabolism roles. We also found specific deletion of CYP2C23s in Feliformia and Mustelidae. This could indicate that eicosanoid metabolism in these animals is different to other carnivorans. A further BLAST analysis indicated possible CYP2C23s orthologues in cattle, horse, and pangolin in loci between Erlin1 and Cpn1 (data not shown), suggesting the CYP2C23 gene translocated after Carnivora divergence. 

### 4.4. CYP3As in Mammals

Human CYP3As catalyzes a wide range of prescribed drugs and is considered one of the most important subfamilies for drug metabolism in humans [2,56], yet they also catalyze metabolism of endogenous chemicals such as steroids, cholesterols, and bile acids [2,57]. Canine CYP3As (e.g., CYP3A12, 3A26, 3A98, and 3A99) have been cloned and characterized, showing similar substrate specificity patterns as human CYP3As [39,58,59]; however, with different expression patterns. In contrast, CYP3A12 and 3A26 are liver specific [58,60], whereas CYP3A98 is expressed in the intestine and 3A99 in the liver and intestine [26]. These features and the phylogenetic result indicate the CYP3As clade 3 could be intestinal-specific isoforms whereas the CYP3As clade 1 could be liver-specific orthologs in Carnivora. However, the CYP3As clade 2 expression patterns remain unclear. This tissue-specific expression of the CYP3As, especially in the liver and intestine, has also been observed in evolutionarily distant species, like humans. 

Among Feline CYP3As, similar expression patterns have been characterized, which means that CYP3A132 is mainly expressed in the liver, whereas CYP131 is mainly expressed in the intestine with lower levels in the liver [27,61,62]. These features, together with our result, further indicate that the Feliformia CYP3A132 may be major CYPs in the liver, whereas CYP3A131 may be intestine specific in Feliformia. In other Feliformia, such as the striped hyena, meerkat, and masked palm civet, however, we found other clades of CYP3As. Unfortunately, Feliformia genomic data are limited and we could not clearly demonstrate the evolutionary history of CYP3As in this taxon. Further studies on a greater variety of Feliformia species’ genomics and expression pattern are necessary. 

Similar to CYP2Cs, we found CYP3As expansion in omnivorous species: the dog, badger, brown bear, and black bear. CYP3As catalyze a wide range of xenobiotics including some pyrrolizidine alkaloids [63,64,65] found mainly in the families Asteraceae, Boraginaceae, and Fabaceae. Thus, the omnivorous diets of bears and badgers might be an evolutionary driving force enabling these species to cope with accidental intake of these toxins. 

Our phylogenetic analysis indicated that diversification of CYP3As in Caniformia and Feliformia was completely different, suggesting that substrate specificity of CYP3As among these taxa is different. However, the phylogenetically distant isoform human CYP3A4 showed similar substrate specificity to CYP3As in dogs and cats (albeit with several differences) [26,27,43], indicating that substrate specificity in Carnivora need not range widely to cope with a variety of chemicals. However, further systemic analysis CYP3As function in Carnivora is required.

### 4.5. Other CYP Isoforms

In our analysis, CYP1-3 families did not show strong differences. However, in domestic cat, specific duplication and loss events were observed. For instance, CYP2C21 has been reported as a pseudogene [66], and CYP2E1, which mainly catalyzes acetaminophen or alcohol, has been specifically duplicated in this species [67]. In our study, we only identified a similar phenomenon of CYP2E duplication in the fishing cat (*Prionailurus bengalensis*) and we did not record CYP2C21 loss in any other Felidae species, suggesting that these duplication and loss events might be specific to cats or Felidae species closely related to cats. Expression of CYP2C has also been limited in cats, whereas CYP2Es are dominant in the liver compared to other isoforms [62,68], which makes it difficult to clarify which isoforms are important for xenobiotic metabolism in Felidae. Thus, a functional analysis is essential for further discussion.

## 5. Conclusions

In this study we comprehensively analyzed the evolutionary features of CYP1-3 families in Carnivora. We found specific expansion of CYP2C and 3As in omnivorous Carnivora such as the badger, the brown bear, the black bear and the dog genomes. Our phylogenetic analysis further revealed possible orthologs of CYP2C21s, 2C41s, and 2C23s in Carnivora. Furthermore, we found the evolution of CYP3As was completely different in Caniformia and Feliformia, and within each taxon we detected specific CYP3As duplication events. These studies provide fundamental evolutionary and genetic information for extrapolating the pharmacokinetics or toxicokinetic of experimental animals to that of wild Carnivora, which include a wide variety of top-predator, key-stone, and rare species threatened with extinction.

## Figures and Tables

**Figure 1 animals-12-02821-f001:**
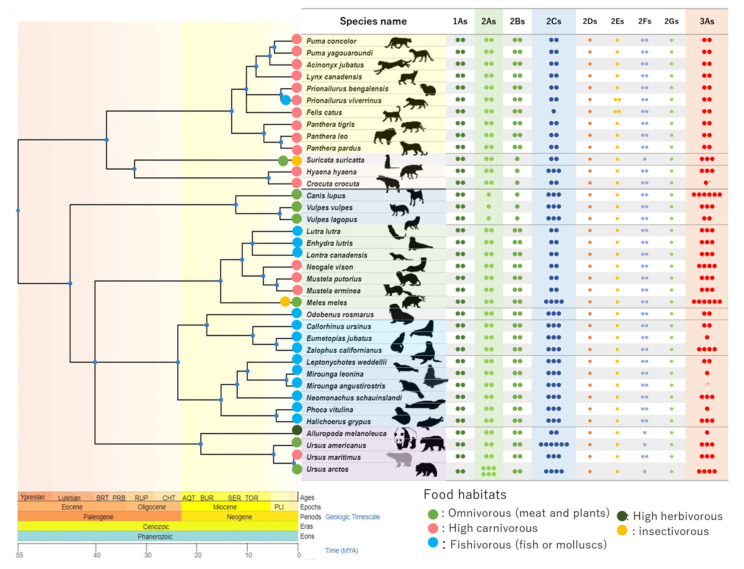
Isoform numbers of CYP1-3 families among Carnivora. Gene numbers for CYP1As, 2A, 2B, 2C, 2D, 2E, 2F, 2G, and 3As are shown by the number of small filled circles. Large filled circles next to the scientific name of each species are colored according to their known diet (Appendix A). Isoforms coding “low quality” or partial genes were not included in this case. The phylogenetic tree was created with TimeTree 5 [23].

**Figure 2 animals-12-02821-f002:**
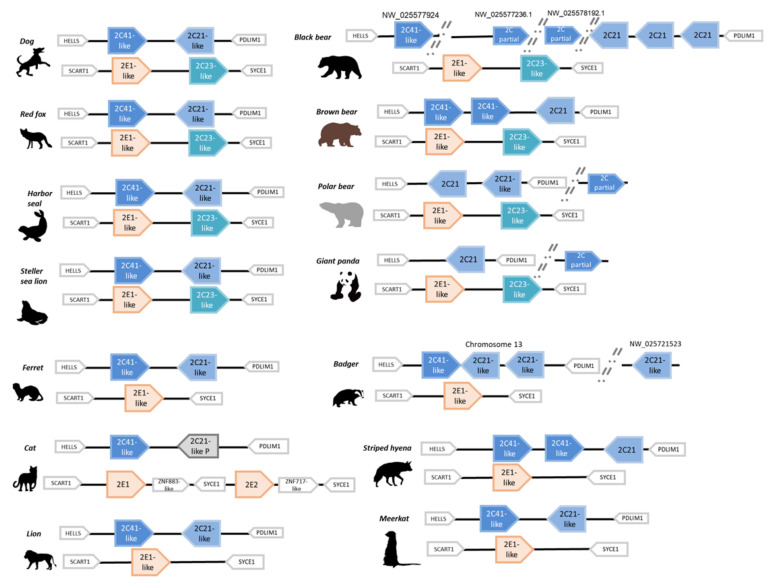
Synteny of CYP2Cs and CYP2CEs clusters in Carnivora. Synteny of the CYP2C cluster and the CYP2CE cluster among Carnivora are shown. Representative species for each family were selected. Phylogenetic-analysis-supported classifications were applied for each isoform and colored based on each clade. CYP2C41s are shown in blue, CYP2C21s in pale blue, CYP2C23s in brilliant blue, and CYP2Es in pale orange. Letters on each locus show coding contigs if separately coded. Pseudogenes are shown as black or gray blocks.

**Figure 3 animals-12-02821-f003:**
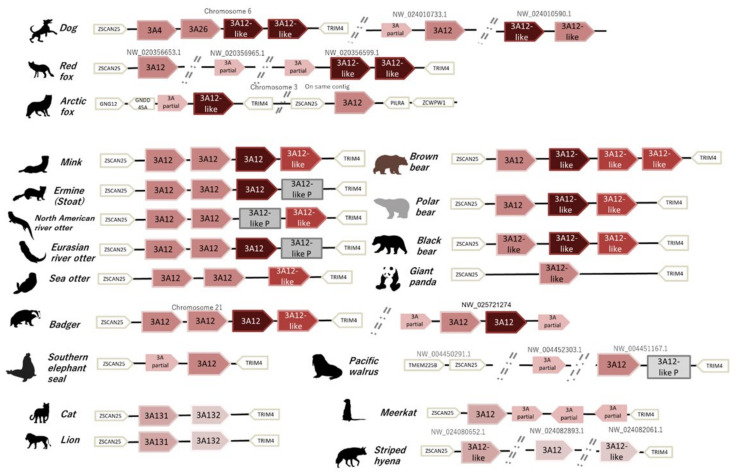
Synteny of the CYP3A cluster among Carnivora. Representative species of each family were selected. Phylogenetic-analysis-supported classifications were applied to each isoform and colored based on each clade. Caniformia CYP3As in clade 1, clade 2, and clade 3, and CYP3A131s and CYP3A132s in Felidae, are all differently colored. Letters on each locus show coding contigs if separately coded. Pseudogenes are shown as gray blocks.

**Figure 4 animals-12-02821-f004:**
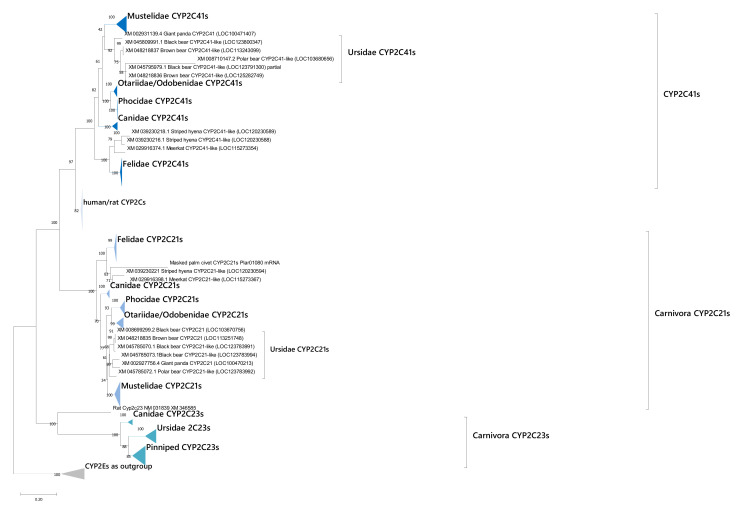
A phylogenetic tree of CYP2Cs sequences in humans and carnivorans. Gene sequences of protein-coding regions for each isozyme were analyzed. The numbers next to the branches indicate the number of occurrences per 100 bootstrap replicates. Genes and clades are tentatively labeled with Carnivoran CYPs examined in this article. Clades of CYP2C41s, CYP2C21s, and Carnivora 2C23s are shown as differently colored triangles. CYP2Es are shown as an outgroup.

**Figure 5 animals-12-02821-f005:**
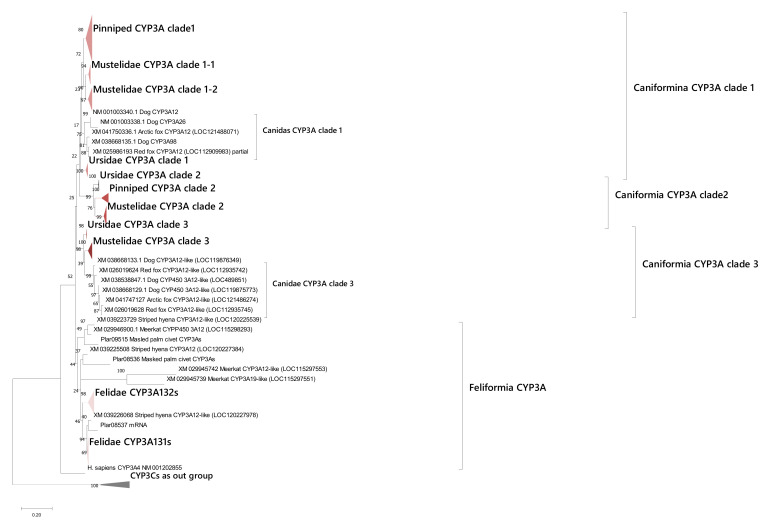
A phylogenetic tree of CYP3As. A phylogenetic tree of CYP3As sequences in carnivorans. Gene sequences of protein-coding regions for each isozyme were analyzed. The numbers next to the branches indicate the number of occurrences per 100 bootstrap replicates. Genes and clades are tentatively labeled with Carnivoran CYPs examined in this article. Caniformia clade 1, clade 2, clade 3, and Felidae CYP3A131s and 3A131s clades are shown with differently colored triangles. CYP3Cs are shown as an out group.

## Data Availability

Not applicable.

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
