# Peer review of "Specific Gene Duplication and Loss of Cytochrome P450 in Families 1-3 in Carnivora (Mammalia, Laurasiatheria)"

_animals, 2022, doi:10.3390/ani12202821_

Round 1

Author Response

Reviewer: 1

The authors analyzed the CYP1-3 genes in various species of the Carnivora to determine their genomic arrangement to understand gene loss and duplication events during evolution. They found the expansion of CYP2C and CYP3A genes in omnivorous species, and differences of the CYP3A evolution between Caniformia and Feliformia. This paper is written well and provides the information that could be the basis for understanding the pharmacokinetic and toxicokinetic profiles of wild Carnivora in the future.

This reviewer has concerns on analysis and presentation of the data.

Some genes presented in the figures were found in the un-scaffolded contigs. Are these genes from the same genome assembly? Because the multiple assemblies are available for some species such as dogs, the same gene might be shown as different gene blocks in the figure. For example, CYP3A12 of the un-scaffolded contig (NW_024010733.1) seems to be nearly identical (99.9%) to known CYP3A12 (shown in Chromosome 6), possibly due to the use of the multiple genome assembly data for the analysis. This reviewer recommends to use one genome assembly data for the analysis in each species. Otherwise, the paper should include the source of the genome assembly data used (e.g. CanFam3 for dogs) and clearly indicate that the genes of the un-scaffolded contigs are not necessarily localized in the same genome as the whole assembled genome. Including this point, this reviewer recommends to describe the limitation of the analysis in Discussion in order to avoid misleading the readers.

Thank you very much for your critical review and comment regarding the possible use of multiple assemblies. We deeply appreciate it.

Especially for Dog (Canis lupus familiaris), we made sure the contigs we utilized in this study was on the same assembly: NCBI reference genome assembly (ROS_Cfam_1.0, GCF_014441545.1). So, we are sure these expanded or duplicated genes in dog were on the same assembly.

Considering your comment, we also added the assemble and annotation information for every species we utilized in this study as Supplemental table S2 on Supplemental Data. Also sentences regarding of this point were changed in L126,

The following latest genome assemblies were used and listed in Supplemental Data.

About the limitation of this analysis, we added the sentence as followed, in the head of Discussion.L281-288.

In this analysis, we utilized the genome assembly data for multiple Carnivorans, and still these data analysis has several limitations. Genome assemble data only represents one individual genomic information and several variants could happen in all genomic assembly. Also, genome assembly qualities should be considered. Although recent genomic assemble quality become improved, and several chromosomal-level assemblies were available for even wildlife Carnivorans, some assembly have relatively low quality which made us difficult to analyze some of CYP genes. Yet, we considered this genomic data analysis have strong importance to evaluate wildlife CYP genome.  

Minor points:

  • Abbreviation needs to be spelled out at the first appearance? (e.g. Line 19, CYP; Line 205, CDS).

We appreciate for your comment, and corrected as followed,

L18: …events of cytochrome P450 (CYP) genes in

L211:  CDS: CoDing Sequence

  • Do species names such as Homo sapiens need to be italicized (Line 83)? Please check throughout the manuscript.

We corrected regarding of this point throughout the maniscript. We appreciate your comment.

  • Ifs better to use the same wording throughout the manuscript, (e.g. Line 104, CYP2C6 blast search -> CYP2C6 BLAST search)

Thank you very much. Wording was corrected as followed,

Blast -> BLAST      : L109, L359

Blastn -> BLASTN         : L111

4) (Line 154) Information on TimeTree 5 needs to be included in Materials & Methods.

The following sentences were added on Materials and Methods, thank you very much

L121:

 The phylogenetic tree was created with TimeTree 5 [22].

5)   Please check typos throughout the manuscript, for example:

> (Line 71)have enable us -> have enabled us

> (Line 82) Data retrive -> Data retrieval

> (Line 214) on same the -> on the same

> (Line 235) CYP3A41s -> CYP2C41s

Thank you very much for your comments and we corrected as follows;

L77: have enabled us

L88: Data retrieval for

L221: on the same

L242: CYP23C41s

We also found other typos and corrected,

L233: CYP3A131s and CYP3A132s

We again deeply appreciate your critical comments and review.

Reviewer 2 Report

Dear Authors:

This is an interesting article describing the effect of lifestyle on P450 profiles concerning food habits. The impact of lifestyle on shaping P450 profiles has been established in microorganisms with too large a sample size. The observation of this study is in contrast to previous comments on the CYP2D gene subfamily (Feng, P. and Liu, Z., 2018. Complex gene expansion of the CYP2D gene subfamily. Ecology and evolution8(22), pp.11022-11030.). However, each gene family and subfamilies are different. To decide on this manuscript, I want to see the protein sequences. Partial mRNAs and transcript variants (which may not be the actual gene since it's predicted) are deduced and may not lead to a change in protein sequences. I request authors provide the real gene numbers, not transcript varvariants, in a table format and protein sequences as a supplementary file. This will undoubtedly enable me to decide on this manuscript.

Author Response

Reviewer: 2

Comments and Suggestions for Authors

Dear Authors:

This is an interesting article describing the effect of lifestyle on P450 profiles concerning food habits. The impact of lifestyle on shaping P450 profiles has been established in microorganisms with too large a sample size. The observation of this study is in contrast to previous comments on the CYP2D gene subfamily (Feng, P. and Liu, Z., 2018. Complex gene expansion of the CYP2D gene subfamily. Ecology and evolution, 8(22), pp.11022-11030.). However, each gene family and subfamilies are different.

  We deeply appreciate your kind and critical comments about this article. Considering your comment above, we added the sentences the part: Introduction  L62-66, mentioning about previous report of CYP2D gene variation and relations with food habitat (Feng, P. and Liu, Z., 2018. Complex gene expansion of the CYP2D gene subfamily. Ecology and evolution, 8(22), pp.11022-11030.).

 Previous report of CYP2D subfamily in mammals showed no clear relation with CYP2D expansion or contractions and food habitat in mammals, suggested some of isoforms didn’t necessarily follow this “plant-arms race” hypothesis [15]. However, this study included only 1 isoform subfamily with wide range of mammals, and evaluation for CYP expansion or contraction in various isoform subfamilies are still needed.

To decide on this manuscript, I want to see the protein sequences. Partial mRNAs and transcript variants (which may not be the actual gene since it's predicted) are deduced and may not lead to a change in protein sequences. I request authors provide the real gene numbers, not transcript variants, in a table format and protein sequences as a supplementary file. This will undoubtedly enable me to decide on this manuscript.

We appreciate your comment, and we added the Supplemental Data of Protein sequence file of CYP2Cs and 3As in FASTA format. We also added the table list of sequences used in Supplemental table S3 and S4. Thank you very much.

Round 2

Reviewer 2 Report

The authors substantially revised the article and addressed the comments. Especially, the new paragraph added at the beginning of the discussion is needed as in the future, some variations might be revealed. Now, I recommend the article for publication.